# Decoding Micromotion in Low-dimensional Latent Spaces from StyleGAN

Qiucheng Wu[1][*], Yifan Jiang[2*], Junru Wu[3*], Kai Wang[5], Gong Zhang[5],
Humphrey Shi[4,5,6], Zhangyang Wang[2], Shiyu Chang[1]
[1]University of California, Santa Barbara, [2]The University of Texas at Austin,
[3]Texas A&M University, [4]UIUC, [5]University of Oregon, [6]Picsart AI Research (PAIR)
qiucheng@ucsb.edu

The disentanglement of StyleGAN latent space has paved the way for realistic and controllable image editing, but does StyleGAN know anything about temporal motion, as it was only trained on static images? To study the motion features in the latent space of StyleGAN, in this paper, we hypothesize and demonstrate that a series of meaningful, natural, and versatile small, local movements (referred to as "micromotion", such as expression, head movement, and aging effect) can be represented in low-rank spaces extracted from the latent space of a conventionally pre-trained StyleGAN-v2 model for face generation, with the guidance of proper "anchors" in the form of either short text or video clips. Starting from one target face image, with the editing direction decoded from the low-rank space, its micromotion features can be represented as simple as an affine transformation over its latent feature. Perhaps more surprisingly, such micromotion subspace, even learned from just single target face, can be painlessly transferred to other unseen face images, even those from vastly different domains (such as oil painting, cartoon, and sculpture faces). It demonstrates that the local feature geometry corresponding to one type of micromotion is aligned across different face subjects, and hence that StyleGAN-v2 is indeed "secretly" aware of the subject-disentangled feature variations caused by that micromotion. As an application, we present various successful examples of applying our low-dimensional micromotion subspace technique to directly and effortlessly manipulate faces. Compared with previous editing methods, our framework shows high robustness, low computational overhead, and impressive domain transferability. Our code is publicly available at https://github.com/wuqiuche/micromotion-StyleGAN.

## 1. Introduction

Recently, the StyleGAN and its variants [1–5] have shown strong performance in controllable image synthesis. These high qualities and fine-grained controls are largely associated with the expressive latent space of StyleGAN. Prior research has revealed that the latent space of StyleGAN is interpretable [6–9], and by manipulating in the latent space, these GANs can generate many images with desired changes [3, 4]. These findings have led to many applications such as face manipulation [10, 11], style transfer [6, 12], image editing [13–15], and even video generation [16–19].

Given this phenomenal result, many try to understand the potential in the latent space of StyleGAN. Particularly, rather than per-image editing methods, people wonder whether it is possible to directly locate latent codes that correspond to sample-agnostic semantically meaningful attributes (*e.g.* smiling, aging on human faces). These attempts can be categorized into supervised and unsupervised methods. The supervised methods [20–22] typically sample a series of latent codes, labeling them with pretrained attributes predictors, and learning classifiers for each desired attribute in the latent space. On the other hand, the unsupervised methods [23, 24] explore the principal components of

---

[*]Equal Contribution.

First Conference on Parsimony and Learning (CPAL 2024).

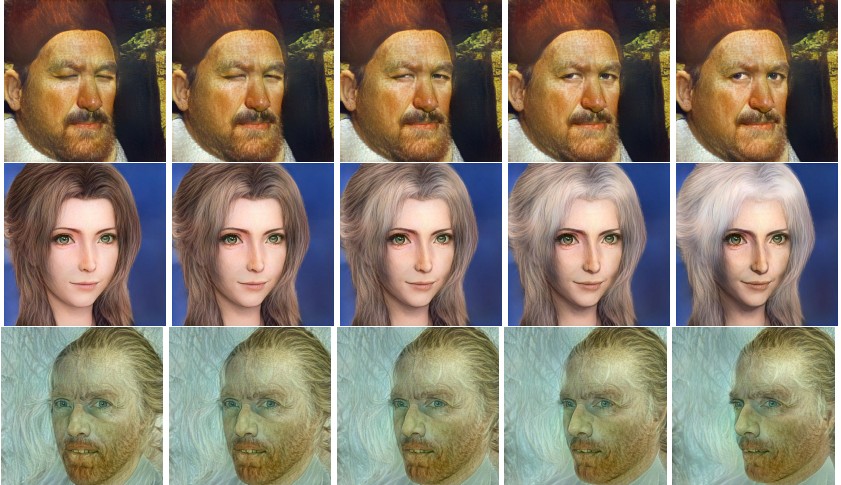

Figure 1: **Representative examples created by the proposed method.** The original images are edited using a simple linear scaling with the discovered universal editing directions on various transformations. These three rows correspond to *eye-opening, aging, and head rotation*.

the sampled latent codes and observe if these codes correspond to semantically meaningful editing directions. However, as will be shown later, the editing directions found by these methods are still entangled with other attributes: When applying these discovered editing directions, the result images suffer from undesired changes in identity and irrelevant attributes. Therefore, we ask: whether this sub-optimal entanglement is due to the intrinsic limits of the entangled latent space, or it is because previous methods do not fully reveal the potentials of the StyleGAN?

To answer the question, we propose in-depth investigations on the StyleGAN-v2's latent space trained on face generations. In particular, we hypothesize that from the StyleGAN's high dimensional latent space, a low-rank feature space can be extracted where universal editing directions can be reconstructed for various facial style transformations including changes in expressions/emotions, head movements, and aging effects, which we refer to as a series of **micromotions**. Thanks to the highly disentangled essence of the decoded editing directions, for any given input, linear scaling along the same found direction will make the image change its style smoothly. Furthermore, to find such a directional vector we leverage the guidance of proper "anchors" in the form of either short texts or a reference video clip and show the directional vector can be efficiently found via simple subtractions using a robustly learned linear subspace projection. Surprisingly, such latent subspace can be extracted using only a single query image, and then the resulting editing direction can be used for any unseen face image, even for those from vastly different domains including oil painting, cartoon, sculpture, *etc*. Figure 1 shows the generated images for multiple style transformations and face types. The contributions of our paper are three-fold:

- We show that in StyleGAN's latent space, there exists low-rank subspaces where universal editing directions can be reconstructed for many meaningful changes (denoted as "micromotions").

- Leveraging a simple framework, we show that these low-dimensional micromotion subspace, along with universal and highly disentangled editing directions, can be consistently discovered.

- As an application, we show that the low-dimensional subspace produces high-quality editing direction, even from vastly different domains (*e.g.*, oil painting, cartoon, and sculpture faces).

## 2. Related Works

### 2.1. StyleGAN: Models and Characteristics

StyleGAN [2–4] is a style-based generator architecture targeting image synthesis tasks. Leveraging a mapping network and affine transformation to render abstract style information, StyleGAN is able

to control the image synthesis in a scale-specific fashion. Particularly, by augmenting the learned feature space and hierarchically feeding latent codes at each layer of the generator architecture, the StyleGAN has demonstrated surprising image synthesis performance with controls from coarse properties to fine-grained characteristics [3]. When trained on a high-resolution facial dataset (*e.g.,* FFHQ [3]), the StyleGAN is able to generate high-quality human faces with good fidelity.

## 2.2. StyleGAN-based Editing

Leveraging the expressive latent space by StyleGAN, recent studies consider interpolating and mixing the latent style codes to achieve specific attribute editing without impairing other attributes (e.g. person identity). [8, 25–28] focus on searching latent space to find latent codes corresponding to global meaningful manipulations, while [29] utilizes semantic segmentation maps to locate and mix certain positions of style codes to achieve editing goals.

To achieve zero-shot and open-vocabulary editing, recent works set their sights on using pretrained multi-modality models as guidance. With the aligned image-text representation learned by CLIP, a few works [10, 30] use text to extract the latent edit directions with textual defined semantic meanings for separate input images. These works focus on extracting latent directions using contrastive CLIP loss to conduct image manipulation tasks such as face editing [10, 30], cars editing [31]. Besides, a few recent works manipulate the images with visual guidance [32, 33]. In these works, image editing is done by inverting the referential images into corresponding latent codes, and interpolating the latent codes to generate mixed-style images. However, these works focus on per-example image editing. In other words, for each individual image input, they have to compute corresponding manipulations in the latent space separately. With the help of disentangled latent space, it is interesting to ask whether we can decode universal latent manipulations and conduct sample-agnostic feature transformations.

## 2.3. Feature Disentanglement in Latent Space of StyleGAN

Feature disentanglement in StyleGAN latent space refers to decomposing latent vector components corresponding to interpretable attributes. Previous studies on StyleGAN latent space disentanglement can be roughly categorized into supervised and unsupervised methods. In supervised methods [20–22], they typically leverage auxiliary classifiers or assessors to find the editing directions. To be more specific, they first sample a series of latent codes from the latent space and render corresponding images. Then, they train an SVM to learn the mapping between sampled latent codes and corresponding attributes, where the labeled attributes are supervised by the auxiliary classifiers. Finally, the normal direction of the hyperplane is the found editing direction. Besides, Goetschalckx et al. [22] directly optimize the editing direction based on an auxiliary classifier. On the other hand, the unsupervised methods [24] typically explore the principal components of sampled latent codes, while they manually check if these components correspond to semantically meaningful attributes. However, as will be shown later, the editing directions found by these methods are shown to be still entangled with other attributes. In this work, leveraging a stronger low-rank latent space hypothesis, we find highly-disentangled latent codes and show that sample-agnostic editing directions can be consistently found in StyleGAN's latent space.

# 3. Method

In this section, we begin by introducing the problem of decoding micromotion in StyleGAN latent space, and we define the notations. Next, in Sec. 3.2, we propose the low-rank micromotion subspace hypothesis, suggesting that the micromotion subspace found from individual entities is consistent across different face subjects. Based on the hypothesis, we demonstrate a simple workflow to decode micromotions and seamlessly apply them to various in-domain and out-domain identities (painting, anime, *etc.*), incurring clear desired facial micromotions.

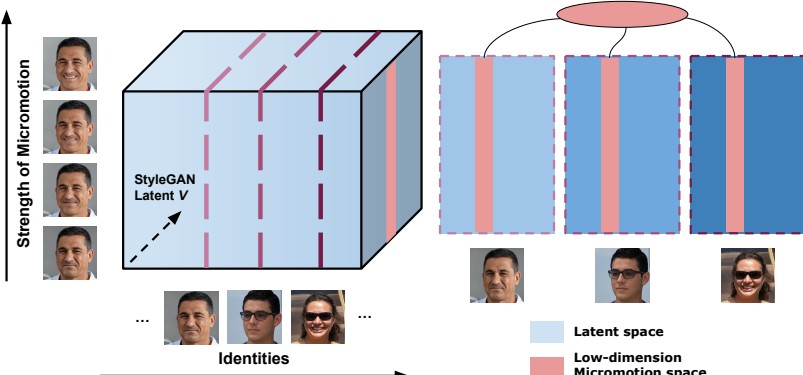

Figure 2: **A tensor illustration of our hypothesis.** In the StyleGAN latent space, we hypothesize the same type of micromotion, *at different quantitative levels but for the same identity*, can be approximated by a low-rank subspace. We further hypothesize that subspaces *for the same type of micromotion found at different identities* are extremely similar to each other, and can hence be transferred across identities.

## 3.1. Problem Setting

Micromotions are reflected as smooth transitions in continuous video frames. Given an input image $I_0$ and a desired micromotion (*e.g.* smile), the goal is to design an identity-agnostic workflow to synthesize temporal frames $\{I_1, I_2, \ldots, I_t\}$, constituting a consecutive video of the micromotion.

Synthesizing images with StyleGAN requires finding proper latent codes in its feature space. We use $G$ and $E$ to denote the pre-trained StyleGAN synthesis network and StyleGAN encoder respectively. Given a latent code $\mathbf{V} \in \mathcal{W}^+$, the pre-trained generator $G$ maps it to the image space by $I = G(\mathbf{V})$. Inversely, the encoder maps the image $I$ back to the latent space $\mathcal{W}^+$, or $\hat{\mathbf{V}} = E(I)$. Leveraging the StyleGAN latent space, finding consecutive video frames turns out to be a task of finding a series of latent codes $\{\mathbf{V}_1, \mathbf{V}_2, \ldots, \mathbf{V}_t\}$ corresponding to the micromotion.

## 3.2. Key Hypothesis: The Low-rank Micromotion Subspace

To generate semantically meaningful and correct micromotions using StyleGAN, the key objective is to find proper latent code series in its feature space. We hypothesize that those latent codes can be decoded by a low-rank micromotion subspace. Specifically, we articulate the key hypothesis in this work, stated as: *The versatile facial style micromotions can be represented as low-rank subspaces within the StyleGAN latent space, and such subspaces are subject-agnostic.*

To give a concrete illustration of the hypothesis, we plot a tensor-view illustration of a micromotion subspace, smile, in Figure 2. The horizontal axis encodes the different face identities, and each perpendicular slice of the vertical plane represents all variations embedded in the StyleGAN latent space for a specific identity. We use the vertical axis to indicate the quantitative strength for a micromotion (*e.g.*, smile from mild to wild). Given a sampled set of images in which a subject face changes from the beginning (e.g., neutral) to the terminal state of a micromotion, each image can be synthesized using a latent code $\mathbf{V}$. Aligning these latent codes for one single subject formulates a *micromotion matrix* with dimension $V \times M$, where $V$ is the dimension of the latent codes and $M$ is the total number of images. Eventually, different subjects could all formulate their micromotion matrices in the same way, yielding a *micromotion tensor*, with dimension $P \times V \times M$ assuming a total of $P$ identities. Our hypothesis is then stated in two folds:

- Each subject's micromotion matrix can be approximated by a subspace and it is inherently low-rank. The micromotion "strengths" can be reduced to linearly scaling along the subspace.

- The micromotion subspaces found at different subjects are substantially similar and mutually transferable. In other words, different subjects (approximately) share the common micromotion subspace. That implies the existence of universal edit direction regardless of identities.

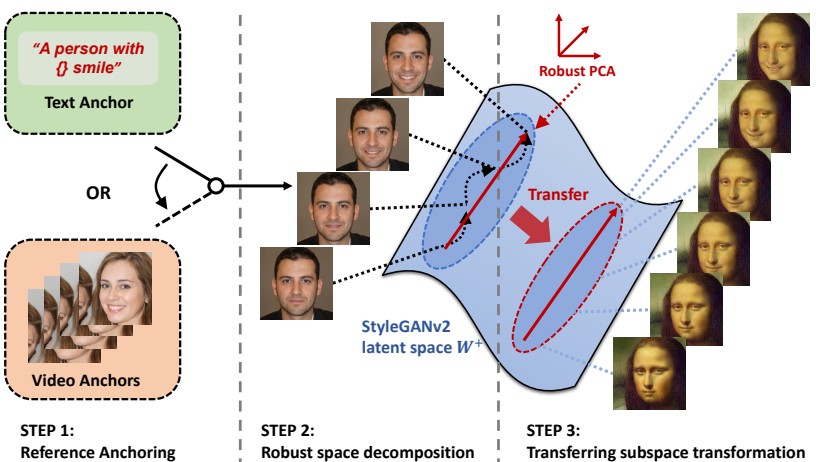

Figure 3: **Our workflow illustration.** In our workflow, we first extract a low-dimensional micromotion subspace from one identity, and then transfer it to a novel identity "Mona-lisa".

If the hypothesis can be proven true, it would be immediately appealing for sample-agnostic image manipulations. First, micromotion can be represented in low-dimensional disentangled spaces, and the dynamic edit direction can be reconstructed once the space is anchored. Second, when the space is found, it can immediately be applied to multiple other identities with extremely low overhead, and is highly controllable through interpolation and extrapolation by as simple as linear scaling.

### 3.3. Our Workflow

With this hypothesis, we design a workflow to extract the edit direction from decomposed low-dimensional micromotion subspace, illustrated in Figure 3. Our complete workflow can be distilled down to three simple steps: (a) collecting anchor latent codes from a single identity; (b) enforcing robustness linear decomposition to obtain a noise-free low-dimensional space; (c) applying the extracted edit direction from low-dimensional space to arbitrary input identities.

**Step 1: Reference Anchoring.** To find the edit direction of a micromotion, we first acquire a set of latent codes corresponding to the desired action performed by the same person. Serving as anchors, these latent codes help to disentangle desired micromotions in later steps. Here, we consider two approaches, text-anchored and video-anchored methods, respectively.

Text-anchored Reference: We follow StyleCLIP [30] to acquire the anchoring latent codes for desired micromotions. The main-idea is to optimize these latent codes by maximizing the embedding similarity between the designed input texts (represent the micromotion) and the images rendered by the codes. Here, one major question is how to design the most appropriate text template to guide the optimization. To generate images with only variance in degrees of micromotions, a natural method is to specify the degrees in the text. For example, for the micromotion "eyes closed", we use both percentages and adjectives to modify the micromotion by specifying "eyes *greatly/slightly* closed" and "eyes *10%/30%* closed". We further study the other text prompts choice in Appendix C.

Video-anchored Reference: Previous method relies on text guidance to optimize the latent codes, while for abstract and complicated motions (*e.g.,* special head movements/ postures), only using text might not be able to express the target micromotion. For this, we leverage a reference video to anchor the subspace instead. Specifically, given a reference video, we invert several of its frames with a pre-trained StyleGAN encoder to obtain the reference latent codes.

After applying either anchoring method, we obtain a set of $t$ referential latent codes denoted as $\{\mathbf{V}_1, \mathbf{V}_2, \ldots, \mathbf{V}_t\}$. We will use these codes to obtain a low-rank micromotion space in later steps.

**Step 2: Robust space decomposition.** Due to the randomness of the optimization and the complexity of image contents (e.g., background distractors), the latent codes from the previous step are

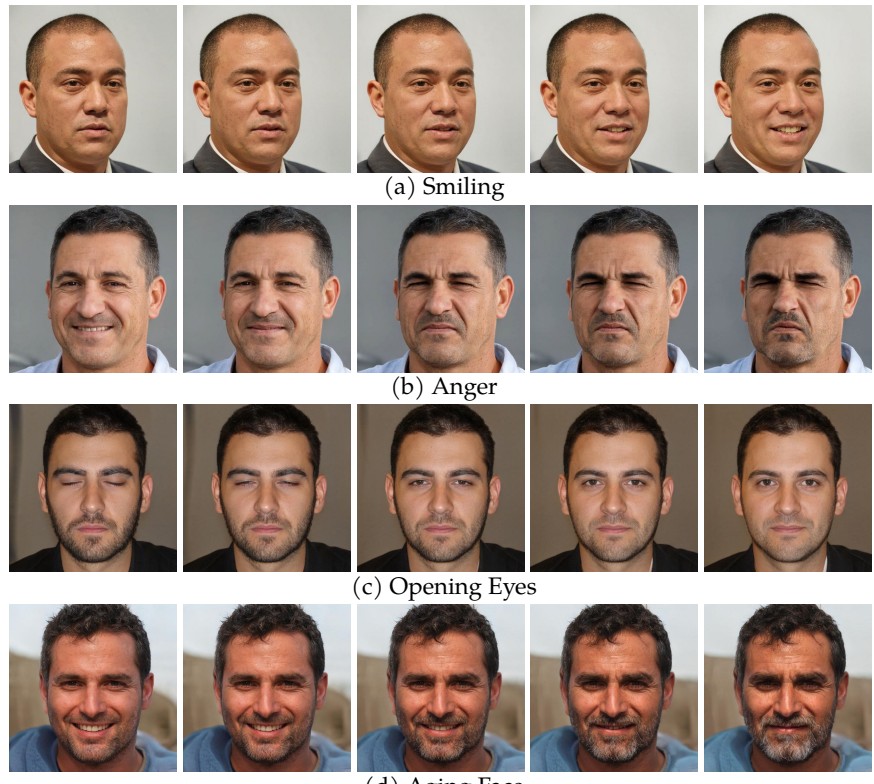

(a) Smiling

(b) Anger

(c) Opening Eyes

(d) Aging Face

Figure 4: **Illustrations of versatile micromotions found by text-anchored method.** We decode the micromotions across different identities, and apply them to in-domain identities. From Top to Bottom: (a) Smiling (b) Anger (c) Opening Eyes (d) Aging Face. Best view when zoomed in.

only "noisy samples" from the underlying low-dimensional space. Therefore, we leverage further decomposition methods to robustify the latent codes and their shared subspace.

The first simple decomposition method we adopt is the principal component analysis (PCA), where each anchoring latent code serves as the row vector of the data matrix. Unfortunately, merely using PCA is insufficient for a noise-free micromotion subspace, since the outliers in latent codes degrade the quality of the extracted space. As such, we further turn to a classical technique called *robust PCA* [34], which can recover the underlining low-rank space from the latent codes with sparse gross corruptions. It can be formulated as a convex minimization of a nuclear norm plus an $\ell_1$ norm and solved efficiently with alternating directions optimization [35]. Through the principal component of the subspace, we get a robust micromotion edit direction $\Delta \mathbf{V}$.

**Step 3: Applying the subspace transformation.** Once the edit direction is obtained, we could edit any arbitrary input faces for the micromotion. The editing is conducted simply through interpolation/ extrapolation along this latent direction to obtain the intermediate frames. For an arbitrary input image $I_0'$, we find its latent code $\mathbf{V}_0' = E(I_0')$, and the videos can be synthesized through

$$I_t = G(\mathbf{V}_t) = G(\mathbf{V}_0 + \alpha t \Delta \mathbf{V}), \tag{1}$$

where $\alpha$ is a parameter controlling the degree of interpolation and extrapolation, $t$ corresponds to the index of the frame, and the resulting set of frames $\{I_t\}$ collectively construct the desired micromotion such as "smiling", "eyes opening". Combining these synthesized frames, we obtain a complete video corresponding to the desired micromotion.

## 4. Experiments

In the experiments, we focus on the following questions related to our hypothesis and workflow:

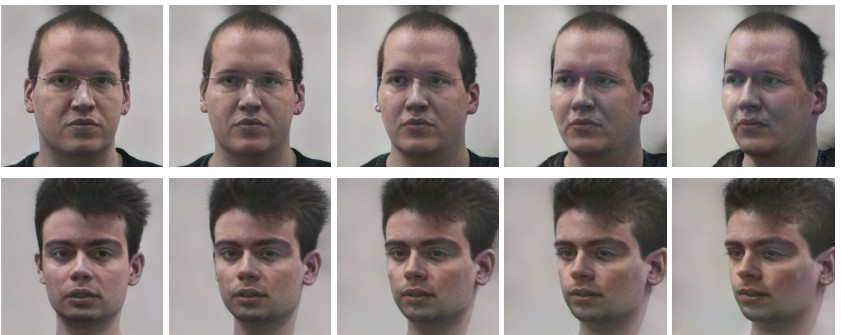

Figure 5: **Illustrations of the micromotion "turning head" found by our video-anchored method.**

- Can we locate subspaces for meaningful and highly disentangled micromotions? (Sec. 4.1)
- Compared with the existing image editing methods, does the located subspace offer higher quality results? (Sec. 4.1)
- Can we painlessly transfer micromotion to other subjects in various domains? (Sec. 4.2)

In short, we target two concepts in following experiments: (a) **Universality**: Our pipeline can consistently find various micromotion, and they can be extended to different subjects across domains; (b) **Lightweight**: Transferring the micromotion only requires a small computation overhead. To validate these concepts, we now analyze our framework by synthesizing micromotions. We mainly consider five micromotions as examples: (a) smiling, (b) angry, (c) opening eyes, (d) turning head, and (e) aging face. We also consider more editings when comparing with other methods.

**Experiment Settings**  The pre-trained models are all loaded from the public repositories [7, 30, 36, 37]. When optimizing latent codes, the learning rate is 0.1 and we use Adam optimizer. For the text-anchored and video-anchored methods, the numbers of latent codes we generate are 16 and 7. In robust PCA, 4 principal dimensions are chosen. More details are in Appendix A.

## 4.1. Micromotion Subspace Decoding

In this section, we use our anchoring methods to locate the micromotion subspace, discovering the editing direction, and apply it to the in-domain identities to generate desired changes. Figure 4 and Figure 5 show the generated five micromotions using text-anchored and video-anchored methods respectively. Within each row, the demonstrated frames are sampled from our synthesized video with the desired micromotions. These results illustrate continuous transitions of human faces performing micromotions, which indicates the edit direction from the micromotion subspace is semantically meaningful and highly disentangled. Therefore, our framework successfully locates subspaces for various micromotions. More analysis and comparison can be found in Appendix B.

### 4.1.1. Quantitative Analysis

To validate if our framework can produce high-quality edits, we compare our decoded micromotions with results from other baselines. We consider InterfaceGAN [20] and GANspace [24]. InterfaceGAN is a supervised method obtaining edit directions from trained SVMs, while GANspace is an unsupervised method that discovers edit directions from the principal components of sampled latent codes. We obtain the editing directions from these baselines respectively, performing edits on 2,000 images, and comparing results quantitatively via the following two analyses.

**Re-scoring Analysis**  First, we quantitatively measure if our discovered editing directions can be successfully disentangled with other irrelevant attributes. Following Shen et al. [20], we perform a re-scoring analysis. Specifically, for a target attribute (*e.g.* smiling), we edit the images using our methods and baselines, and we use the scores from pretrained classifiers [38] to measure how the edits influence the target attribute as well as non-target attributes. Ideally, a well-disentangled edit should result in a major change for the target attributes, with minor influence on others.

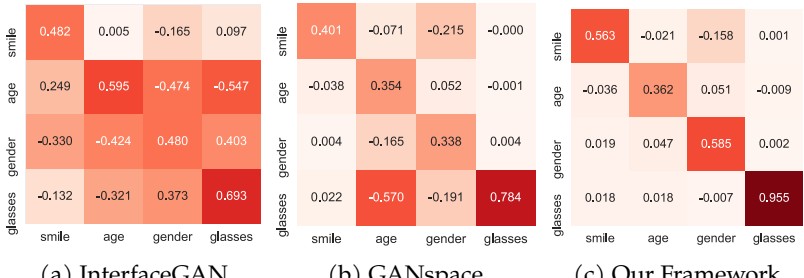

|  (a) InterfaceGAN | (b) GANspace | (c) Our Framework |

Figure 6: **Re-scoring analysis.** In each row, we perform editing targeting the attribute of the row, and each column shows the corresponding changes scored by different pre-trained attribute classifiers, respectively. Higher scores indicate larger changes. We notice a clear diagonal pattern when editing latent space using our framework, indicating its strong disentangle property.

Our results[2] are shown in Figure 6. In these three confusion matrices, for each row, we run latent-space editing with the target attribute, and each column shows the corresponding changes measured by pretrained attribute classifiers. For example, in the matrix of InterfaceGAN, the first row means when editing with target attribute "smile" and measuring the changes using pretrained classifiers, the score of "smile" increases by 0.482 while the score of "age" also increases by 0.005, *etc.* Higher scores indicate larger changes. We observe that our framework produces a clear diagonal pattern, while the baseline methods lead to significant changes in both target and non-target attributes. Therefore, our framework demonstrates stronger disentangle performance.

**Identity-agnostic Analysis** Preserving the identity is essential in face editing. Following [20], we perform an identity-agnostic analysis, where we use a pre-trained face identifier [38] to quantitatively evaluate if the identity is changed after edits.

Table 1: Identity-agnostic Analysis. Each row shows the changes in identity score (the smaller the better) when editing the attributes in latent space.

|  | Smile | Age | Gender | Glass |
|---|---|---|---|---|
| InterfaceGAN | 0.0515 | 0.1294 | 0.1225 | 0.0916 |
| GANspace | 0.1081 | 0.0975 | 0.0507 | 0.1420 |
| Ours | **0.0047** | **0.0660** | **0.0491** | **0.0279** |

We compare our framework with other editing baselines and demonstrate the results in Table 1. Here, each row shows the changes in identity score when editing the target attributes, and a smaller score indicates better identity preservation. In this table, we find our method preserves identities well in different edits, while Other baselines incur more severe identity changes. Therefore, we find that our framework is able to discover more precise editing directions and better disentangles the target attributes from the human identities.

## 4.2. Micromotion Applications on Cross-domain Identities

Next, we further explore if decoded micromotions can be applied to cross-domain identities. Figure 7 shows the result of transferring the decoded micromotions on novel identities. Within each row, we exert the decoded micromotions on the novel identities, synthesize the desired movements, and demonstrate sampled frames from the generated continuous videos. From these results, we observe that the sampled frames on each new identity also depict the continuous transitions of desired micromotions. This verifies that the decoded micromotions extracted from our workflow can be successfully transited to the out-domain identities, generating smooth and natural transformations. Also, this shows the low-dimensional micromotion subspace in StyleGAN is indeed not isolated nor tied to certain identities. In fact, the *identity-agnostic* micromotions can be found using our framework and can be ubiquitously applied to those even out-of-domain identities.

Moreover, we emphasize that to generate micromotion on a novel identity, the entire computational cost boils down to inverting the identity into latent space and extrapolating along the edit direction, without the requirement of conducting identity-specific computations. This leads to effortless editing of new identity images using the found direction, with little extra cost.

---

[2]Since the classifiers used by Shen et al. [20] are not available, we use different classifiers [38].

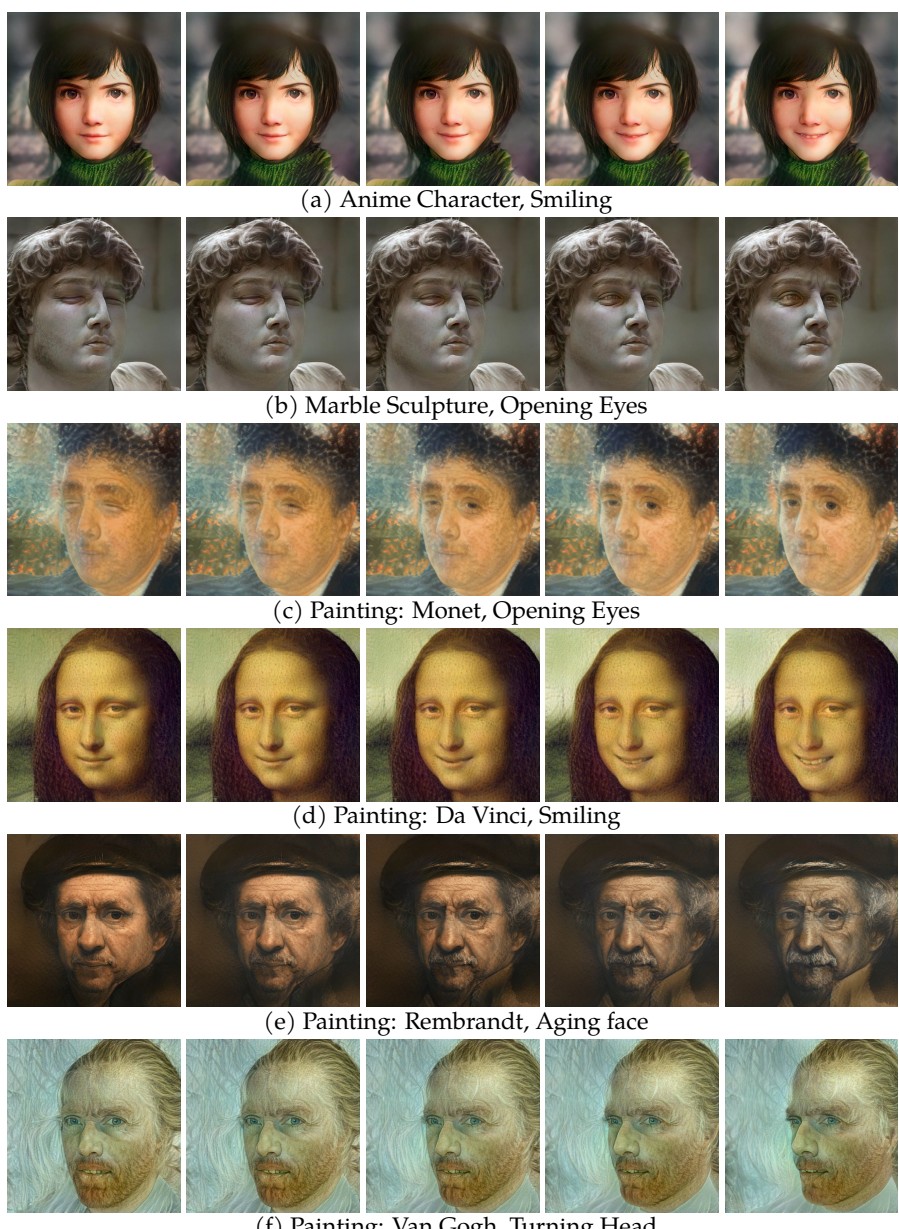

(a) Anime Character, Smiling

(b) Marble Sculpture, Opening Eyes

(c) Painting: Monet, Opening Eyes

(d) Painting: Da Vinci, Smiling

(e) Painting: Rembrandt, Aging face

(f) Painting: Van Gogh, Turning Head

Figure 7: **Micromotions on cross-domain identities.** Our micromotions generalize well when transferred to novel domains, including anime characters, sculptures, and various genres of paintings.

# 5. Conclusions

In this work, we analyze the latent space of StyleGAN-v2, demonstrating that although trained with static images, the StyleGAN still captures temporal micromotion representation in its feature space. We find versatile micromotions can be represented by low-dimensional subspaces of the original StyleGAN latent space, and such representations are disentangled and agnostic to the choice of identities. Based on this finding, we explore and successfully decode representative micromotion subspace by two methods: text-anchored and video-anchored reference generation, and these micromotions can be applied to arbitrary cross-domain subjects, even for the virtual figures including oil paintings, sculptures, and anime characters.

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

## A. Implementation Details

For text-anchored experiments, the original images are generated using random latent codes in StyleGAN-v2 latent space. The text prompts are in the general form of (a) "A person with {} smile"; (b) "A person with {} angry face"; (c) "A person with eyes {} closed"; (d) "{} old person with gray hair", which correspond to the micromotions of smiling, angry, eyes opening and face aging. Here, the wildcard "{}" are replaced by a combination of both qualitative adjectives set including {"no", "a big", "big", "a slight", "slight", "a large", "large", " "} and quantitative percentages set including {10%, ..., 90%, 100%}. We will discuss the choice of various text templates and their outcomes in the ablation study. For the video-anchored experiments, we consider the micromotion of turning heads. The referential frames are collected from the Pointing04 DB dataset [39], and the frames we used for anchoring include a single identity with different postures, which has the angle of $\{-45°, -30°, -15°, 0°, 15°, 30°, 45°\}$.

## B. Qualitative comparison with baselines

In this experiment, we qualitatively compare the performance of our method with existing editing methods. We consider the following baselines: InterfaceGAN [26], GANspace [24], StyleFlow [40], [41], and MoCoGAN-HD [42].

**Experiment Setting.** All methods are tested on the StyleGAN-v2 pretrained on the FFHQ dataset, and we follow the settings stated in Sec. 4. We adopt the released pretrained models for most of the baselines except Zhuang et al. and StyleFlow. For Zhuang et al., since it is only trained on $256 \times 256$ images, we first attempt to train the model on $1024 \times 1024$ using the code released by authors for a fair comparison. However, the model does not converge on 1024 resolution. Therefore, we perform comparisons by collecting the source and edited images shown on their papers and using our method to perform the edit. For StyleFlow, since the code is in Tensorflow 1.x and not supported by our machines, we also compare with the images on their papers qualitatively. For most baselines, we compare their editing results on two representative target attributes in the paper: "smiling" and "aging". We choose these two attributes because these are the common attributes explored by both our method and baselines. Finally, for MoCoGAN-HD, since it is only trained on the "talking-head" task on FFHQ dataset, we compare the editing performance on this task. Specifically, we apply our video-anchored method to synthesize videos of a person talking using a single input reference video. Meanwhile, videos of MoCoGAN-HD are synthesized using their pretrained models.

**Results.** We first demonstrate the comparison between our method and InterfaceGAN, GANspace. The comparison result is shown in Figure 8. From the result, we observe that our method demonstrates comparable or better performance than existing baselines. Specifically, compared with these methods, our method faithfully preserves the identities of the edited subjects, while InterfaceGAN often changes the identities, even genders, during edition, and GANspace usually produces minor edit towards target attributes. Also, to see the benefits of on-demand disentanglement vs. manually picking from principal components, we demonstrate the first 14 principal directions from GANspace in Figure 11. We observe that (1) all of these latent codes cannot open/close people's eyes, meaning GANspace fails to find editing directions for this attribute even after manually checking 14 directions in this case; (2) some of the editing directions are clustered, e.g, attribute "glasses" is entangled with "age" in C3, and is entangled with "gender" in C9. On the other hand, our method can find editing directions for these attributes without heuristic human choices.

Besides, we demonstrate the comparison between our method and MoCoGAN-HD in Figure 9. In the figure, the first six rows show synthesized videos by our methods. Starting from a single input video (1st and 4th row), our video-anchored method inverts their frames, reconstructs the original video (2nd and 5th row) and produces the same talking actions on a new identity (3rd and 6th row). The last two rows show the result of MoCoGAN-HD. From the results, we highlight two benefits of our methods: First, videos synthesized by our method has more significant variance than MoCoGAN-HD. Our synthesized videos show a person talking with mouth and eyes actions, while most frames in MoCoGAN-HD resemble the first frame and have little changes. Second, our method allows on-demand talking actions, *i.e.*, the synthesized video resembles the reference video at each frame. On the other hand, MoCoGAN-HD cannot control the synthesized talking action. Besides these two benefits, we emphasize that our method only requires a single input video to generate talking motions on novel identity, while MoCoGAN-HD requires a large training dataset (*e.g.*, VoxCeleb [43] dataset with 22,496 clips). With these advantages, we conclude that our method is more effective and convenient than MoCoGAN-HD in the talking-head task.

Finally, the comparison between our method and StyleFlow, Zhuang et al. are shown in Figure 10[3]. We observe that our method and two baseline methods result in different styles of "smile" and "aging", while the quality is comparable. Besides, we emphasize that both StyleFlow and Zhuang et al. require training auxiliary models, while our method does not need any new models, and finding editing direction using our method can be done in a few minutes. Therefore, our method outperforms the listed baselines by providing a more convenient editing framework.

## C. Ablation Studies and Explorations

In this section, we perform a series of ablation studies and explore several design choices in our proposed framework.

**Changing identities of anchoring latent codes**   In this ablation, we explore if the choice of identity influences the micromotion quality. We use photos of different people (denoted as Identity A, B, and C) to discover editing directions, and we generalize to the same sketch painting. The result is shown in Figure 12. We observe that latent codes decoded from various identities generate visually similar micromotions. Therefore, the micromotion can be decoded using different identities and still result in semantically correct edits.

**Ablation on subspace decomposition techniques**   In this ablation study, we compare different subspace decomposition techniques, including Robust PCA, Vanilla PCA, and without PCA. In Figure 13, we show Robust PCA yields the best visual results, followed by Vanilla PCA, while without PCA yields results with the worst visual quality. When comparing the results using vanilla PCA with robust PCA, we can observe the former creates more undesired artifacts. For example, in the third column of Figure 13, we observe vanilla PCA create an unwanted artifact around the shoulder of the sculpture, while robust PCA provides a cleaner image. On the other hand, micromotion subspace without PCA decomposition creates images with the worst quality. Most of them have serve distortion and the faces are barely recognizable. The ablation demonstrates vanilla PCA is insufficient for a noise-free micromotion subspace, while the Robust PCA is a more favorable choice.

**The role of text templates**   To explore the sensitivity of the micromotion subspace w.r.t the text templates, we study various text templates that describe the same micromotion. In Figure 14 top row, we can see that the micromotion "closing eyes" is agnostic to the choice of different text templates and generate similar visual results. On the other hand, In Figure 14 bottom row, we observe the opposite where the micromotion "face aging" is sensitive to different text templates, which results in diverse visual patterns. This suggests the choice of text template may influence the performance of some micromotions, and a high-quality text guidance based on prompts engineering or prompts learning could be interesting future work.

---

[3]The original images used in our work are inverted from the corresponding input images for baselines. Therefore, the original images are slightly different.

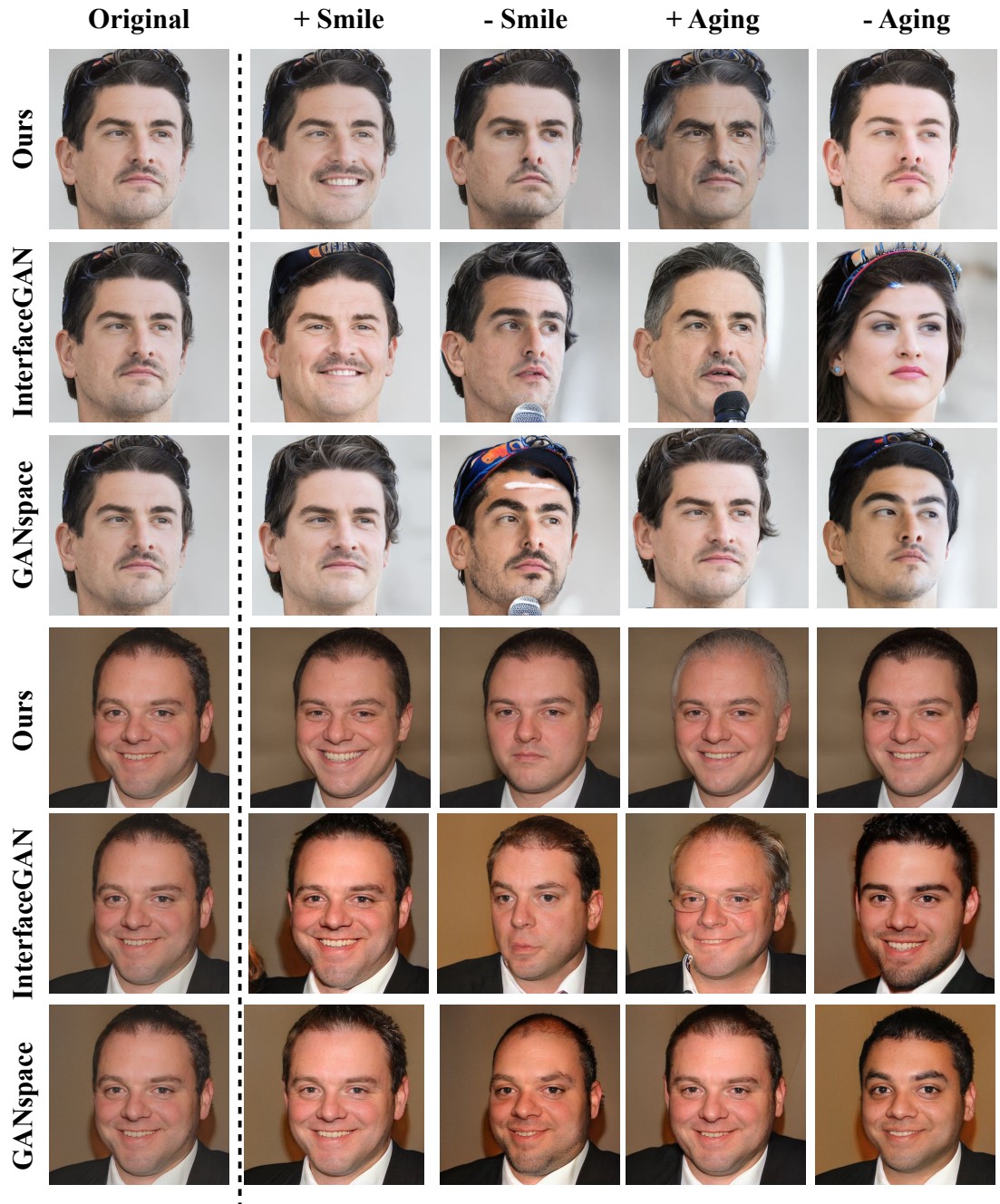

Figure 8: **Qualitative comparison with InterfaceGAN and GANspace.** The target attributes are "smiling", "aging" for human faces.

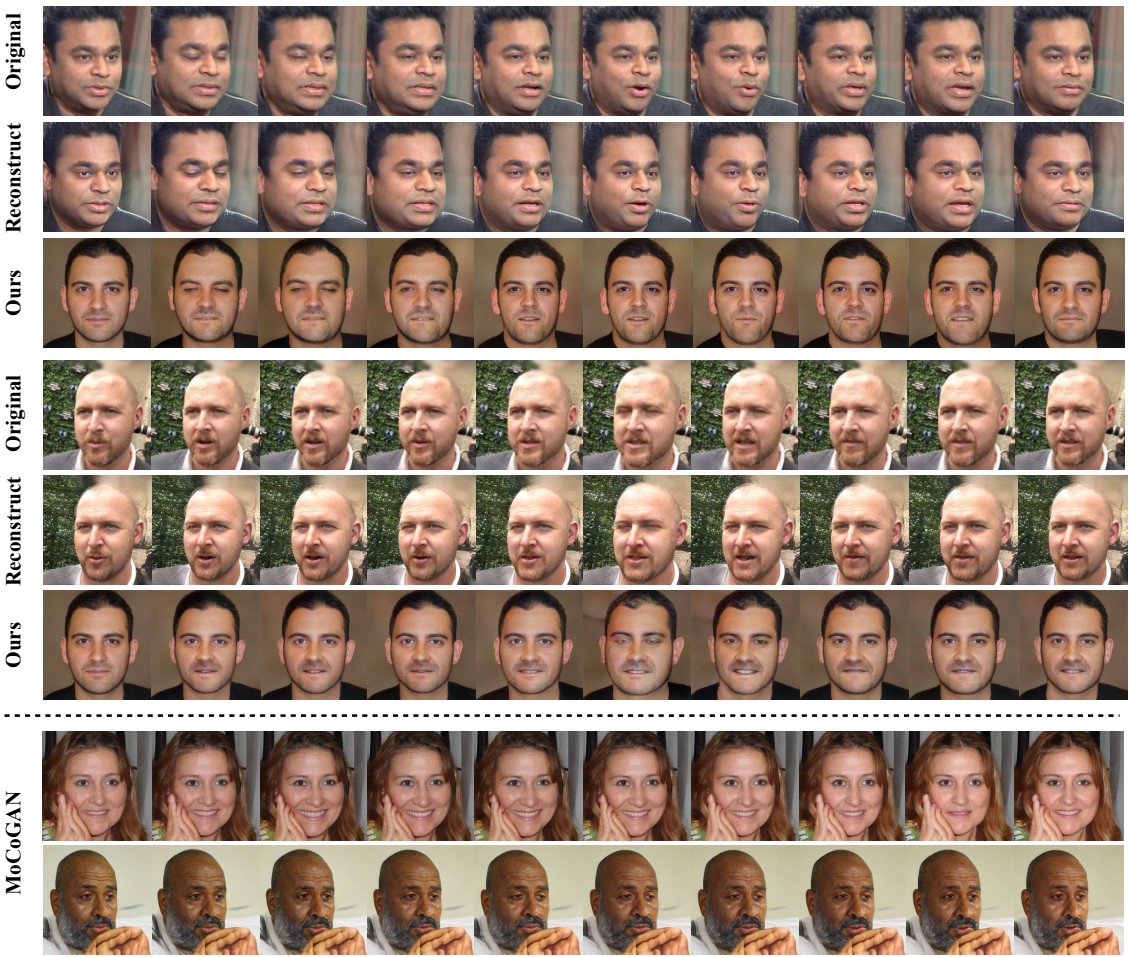

Figure 9: **Qualitative comparison with MoCoGAN-HD on task "talking head".** The first six rows show synthesized videos by our methods. Starting from single input video (1st and 4th row), our video-anchored method invert their frames, reconstruct the original video (2nd and 5th row) and produce the same talking actions on new identity (3rd and 6th row). The last two rows show the result of MoCoGAN-HD.

**Changing the number of anchors and identities** In our framework, we rely on a series of latent codes to anchor the low-rank space, and these codes are obtained from one identity performing micromotion. Therefore, we further ask two questions for our design: (a) How would the number of latent codes influence the editing performance? (b) Could we obtain better editing results from multiple identities? For the first question, we hypothesize the number of latent codes would influence the quality of the discovered low-dimensional space, therefore influencing the editing performance. For the second question, although we have obtained high-quality micromotion editing direction from a single identity, we explore if multiple identities decrease the correlation between micromotion and identity and lead to better disentanglement.

We use different numbers of anchoring latent codes and identities to discover the editing direction and apply it to novel images. In the first study, multiple latent codes are used to determine the low-rank space. Notice that when using only one anchoring latent code, the framework reduces to using StyleCLIP to find editing direction and directly apply it to novel images. In the second study, with multiple identities, we optimize the latent code on each identity separately and use the average latent code as the final editing direction.

|  | Original | + Smile | Original | + Smile | Original | +Age |
|---|---|---|---|---|---|---|

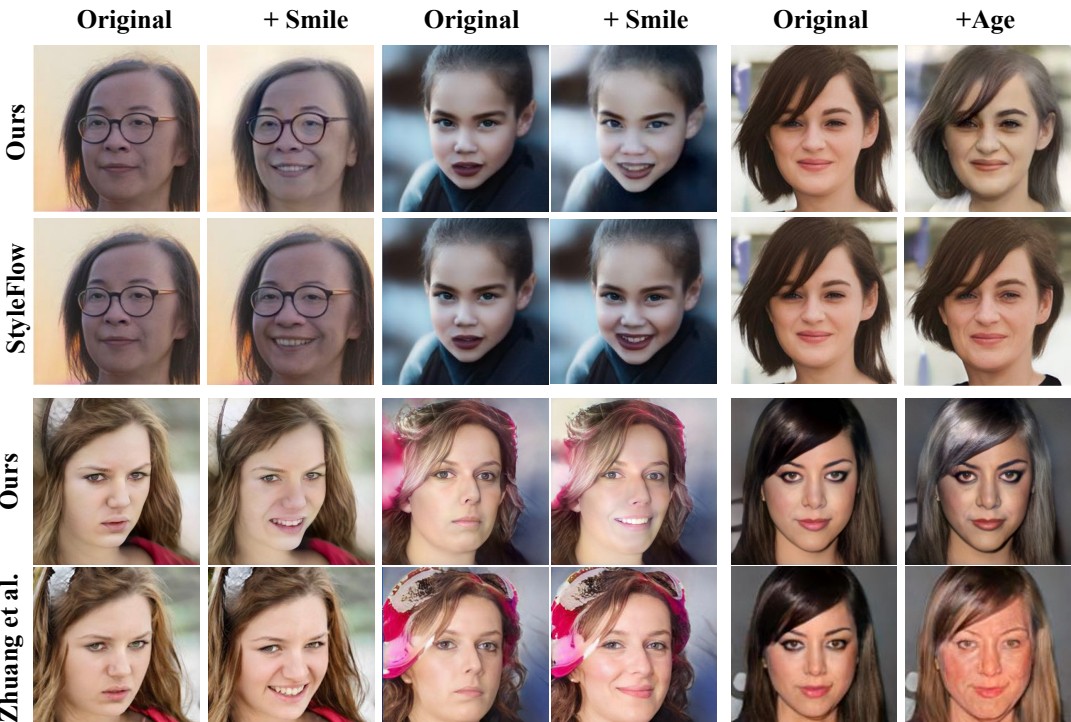

Figure 10: **Qualitative comparison with StyleFlow and Zhuang et al.**

Our results can be found in Fig 15 and Fig 16. For the effects of anchoring latent codes, we observe that fewer anchors result in noises and artifacts, indicating insufficient disentanglement. Meanwhile, we observe the quality of latent space editing improves gradually with respect to the number of anchors. This motivates us to use a series of anchoring latent codes for a better low-rank latent space. For the effects of identities, we observe that although using more identities has some weak benefits (*e.g.* better background preservation), there is no clear visual improvement compared with using one identity. This motivates us to stick with one identity with better efficiency in our framework.

**Effects of using different GANs** Besides the StyleGANv2 discussed in main paper, in this study, we further discuss if we can disentangle from progressive GAN [1] and BigGAN [44].

For both progressive GAN and BigGAN, we adopt the publicly released pretrained models. The progressiveGAN is loaded from pytorch hub. The BigGAN is loaded from the original repository. For the editing tasks, we choose the target attributes according to their training datasets. Specifically, for progressive GAN, the model is trained on CelebA [45] dataset, and we study the attributes "smiling" and "aging" on human faces. For BigGAN, the model is trained on ImageNet [46] dataset, we study the attributes "opening mouth" and "closing eyes" on dogs. We use the same text prompts for the human face experiment ("A person with {} smile", "{} old person with gray hair"). For the experiment on dogs, the text prompts we construct are in a similar form ("A dog with eyes {} close", "A dog with mouth {} open").

The result is shown in Figure 17. From the figure, we observe that both GANs do not synthesize high-quality editing images. For example, for the "opening mouth" attribute in BigGAN, the mouths of dogs in the first two rows are larger, but both the dogs and backgrounds change drastically. This is even worse for the target attribute "closing eyes". Similarly, in Progressive GAN, we find slight changes toward target attributes "smiling" and "aging", while the identities are largely changed. This result indicates the latent space in BigGAN and progressive GAN are not highly disentangled. There are two possible reasons: First, the latent code dimension in BigGAN and Progressive GAN ($1 \times 512$) is smaller than the one in StyleGANv2 ($18 \times 512$). Second, the hierarchical

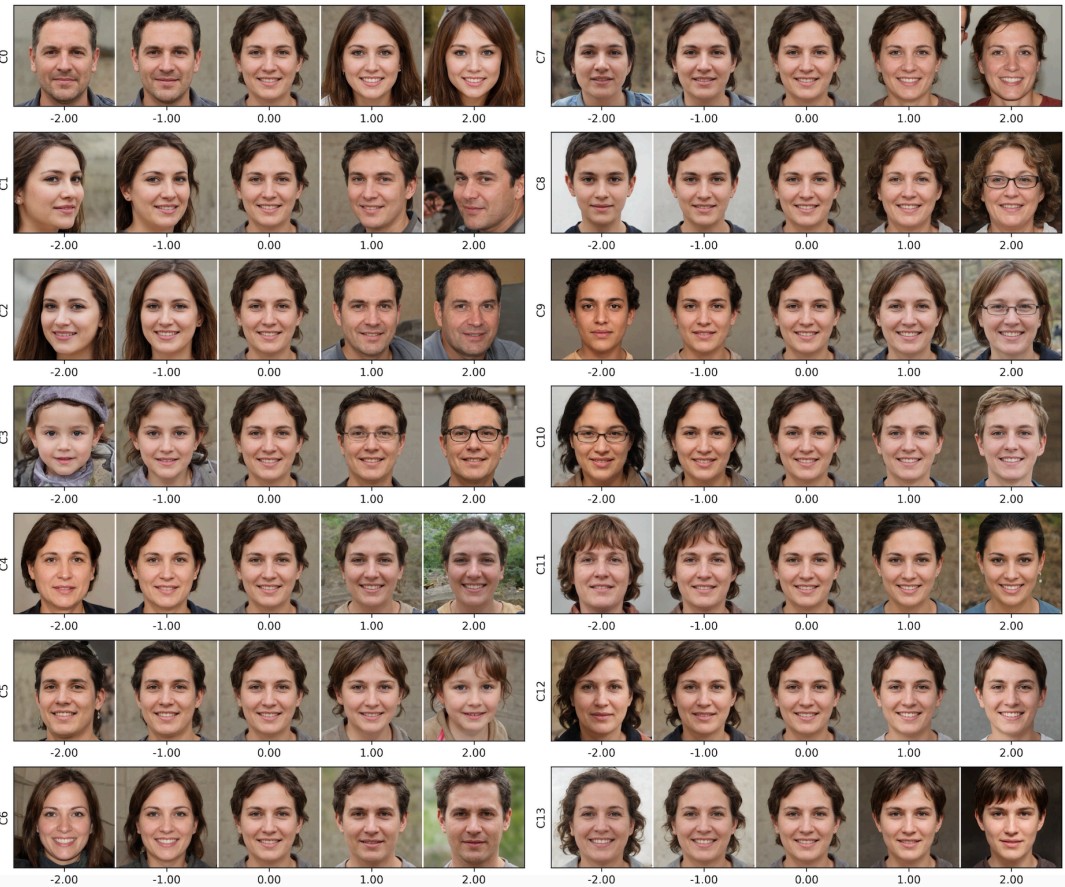

Figure 11: **Principal Components of GANspace.** We demonstrate the editing effects of using the first 14 principal components to edit images.

structure in StyleGAN might lead to better disentanglement. Therefore, compared with these generators, the StyleGANv2 used in the main paper is a better choice.

**Changing inversion methods**  In this experiment, we demonstrate how different inversion methods influence editing performance. We consider three methods: Restyle+pSp [36, 47], Restyle+e4e [48], and vanilla e4e method. The result is demonstrated in Figure 18. We observe that using different inversion methods influences the editing results. Arguably, Restyle+pSp preserves the background color and details best. Besides, we also observe that other methods produce undesired changes (*e.g.*, images are darker for the upper rows). Choosing a faithful inversion method helps produce high-quality edits.

## D.  Additional examples of micromotions transferred to novel domains

In Figure 19, we include additional visual examples to demonstrate that our micromotions generalize well when transferred to novel domains. The additional novel domains include bronze sculptures, oil/sketch painting, and more anime characters.

## E.  Failure case

While our method can disentangle many micromotions and transfer to novel images in different fields, we would like to demonstrate a few limitations of our framework. First, the editing ability

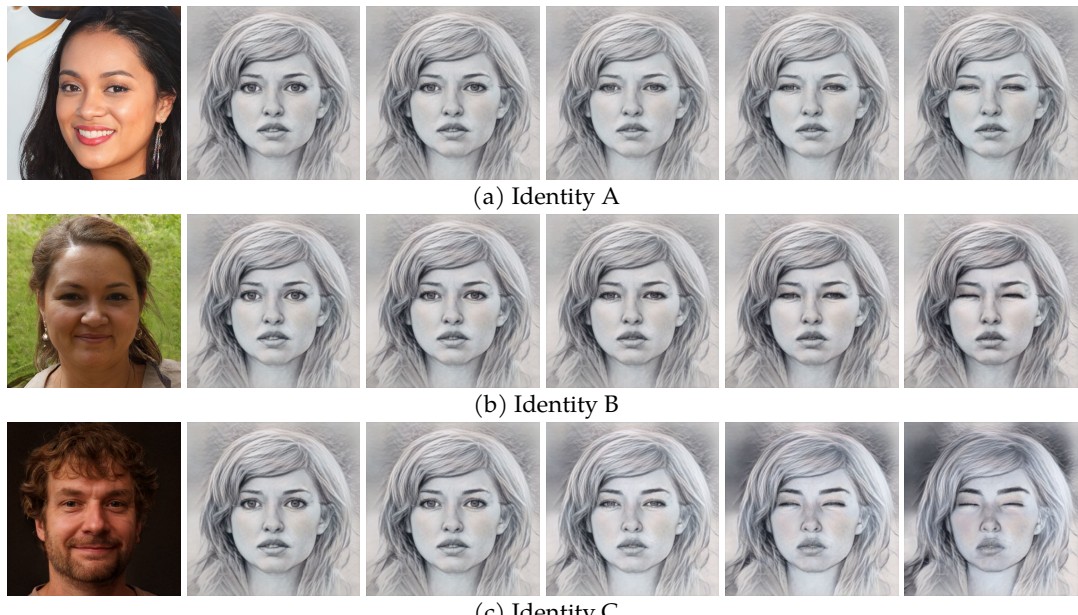

(a) Identity A

(b) Identity B

(c) Identity C

Figure 12: **Effects of changing the identities of anchoring latent codes.** The first column shows synthesized images of different identities generated by three latent codes. From the second column, we show the micromotion subspace (i.e. "closing eyes") decoded from those three identities, exhibiting visually similar results, when generalized to a sketch painting in the novel domain.

of our method originates from the disentangled latent space of StyleGANv2. We have shown in section C that, with a less disentangled GAN architecture, this framework cannot produce high-quality editing results. Second, when transferring the editing directions to out-of-domain images, we first need to invert the input images to vectors in the latent space of StyleGAN. When the input largely deviates from a photo-realistic person, the inversion model fails to find the corresponding latent code, and therefore the editing will also fail. We provide an example in Figure 20. Here, the target attribute is "smiling", and the input images are anime characters (Mario, Chihiro). We find that the latent code produced by the encoder cannot reconstruct the images (in the second and fifth row), and therefore the editing images have poor quality.

## F. Micromotions other than human face

Finally, we provide more micromotion examples on subjects other than the human face. In this experiment, we explore the micromotions of wild animals. The StyleGANv2 model we used is pretrained on the AFHQ-wild dataset [49] with $512 \times 512$ resolution, and we consider "eyes close" and "mouth close" as two representative micromotion examples. The results can be found in Figure 21. From the figure, we observe that our method can also synthesize micromotions on wild animals, while the quality is not as good as those on human faces. We highlight two drawbacks here. First, the synthesized images change the background as well. Second, the synthesized images sometimes do not reflect a smooth micromotion. We provide one example in the first row. Specifically, we expect the wild animal to gradually close its eyes, while the synthesized images demonstrate a pixel-wise interpolation from open eyes to close eyes. We hypothesize this is due to the AFHQ-wild dataset does not contain wild animals with different eyes open degrees. As such, interpolation on the editing direction cannot synthesize animals with eyes half-open, which is hardly seen in the training dataset. We believe that with a high-quality dataset and better-pretrained generator, we can expect better micromotions.

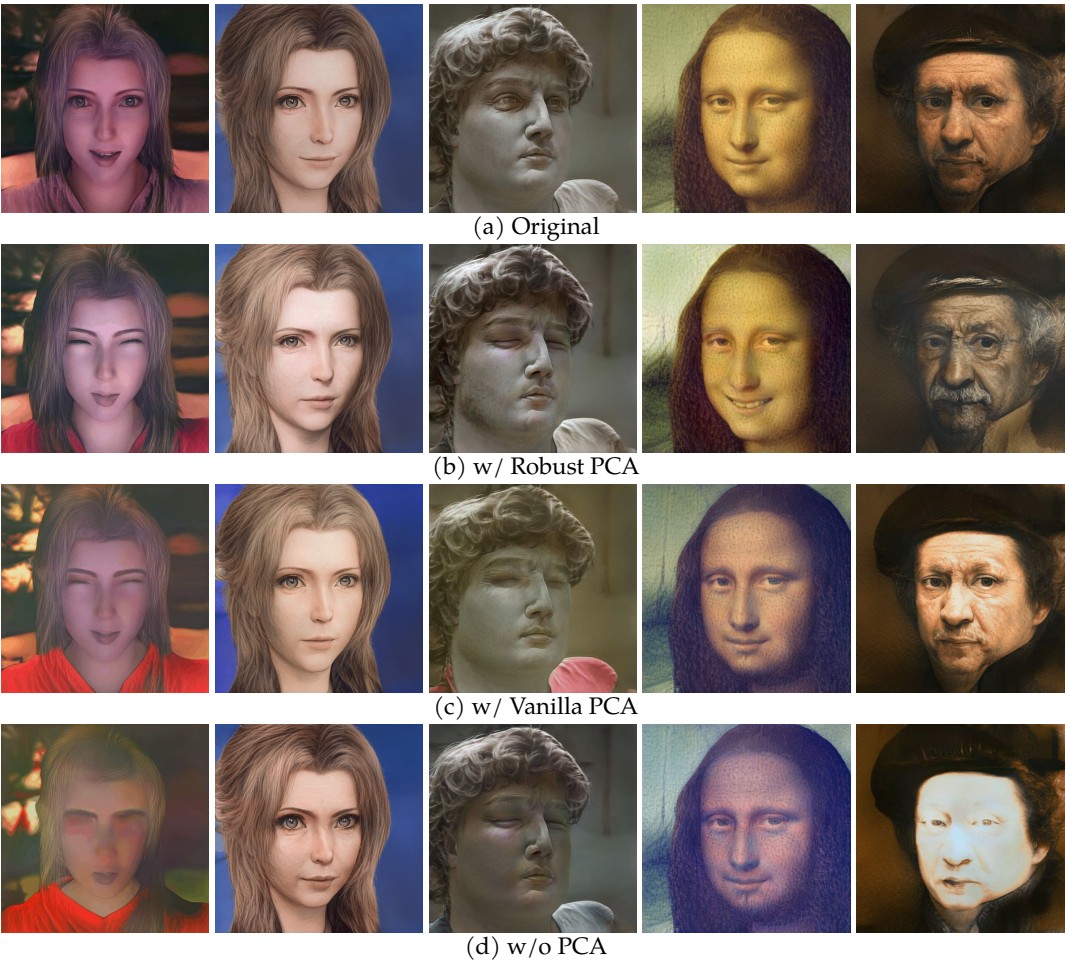

Figure 13: **Comparison between Vanilla PCA, Robust PCA and without PCA.** For each column, from left to right, the micromotions are "closing eyes" (for the first and third columns), "angry", "smiling", and "aging face". For conciseness, we only show the original and last frame. Best view when zoomed in.

# G. Performance on other StyleGANs

In the previous section, our primary focus was on StyleGAN-v2, a prominent representative within the StyleGAN family of architectures. In this section, we aim to broaden our investigation by assessing whether our discoveries can be extrapolated to other advanced StyleGAN models. Specifically, we perform experiments on StyleGAN-v3 [2] and StyleGAN-XL [5], which are two recent StyleGAN models with larger architecture and better general performance. We utilize publicly accessible pretrained models on the FFHQ dataset for both StyleGAN-v3 and StyleGAN-XL. For the editing tasks, we use the example attributes "smile" and "angry" following our previous experiments. We also use the same text prompts for these experiments and follow the same procedures to find and apply subspaces.

The experiment result is demonstrated in Figure 22 and Figure 23. From these results, we observe the micromotion subspaces consistently emerge within both of these architectural frameworks, yielding fluid and semantically meaningful transitions. However, it is worth noting the occasional discrepancies in these transitions, exemplified by background alterations in the final row of Figure 22 and Figure 23. These issues can be attributed to two primary factors: First, although with robust-PCA, since we are only using several latent codes as reference images, the discovered sub-

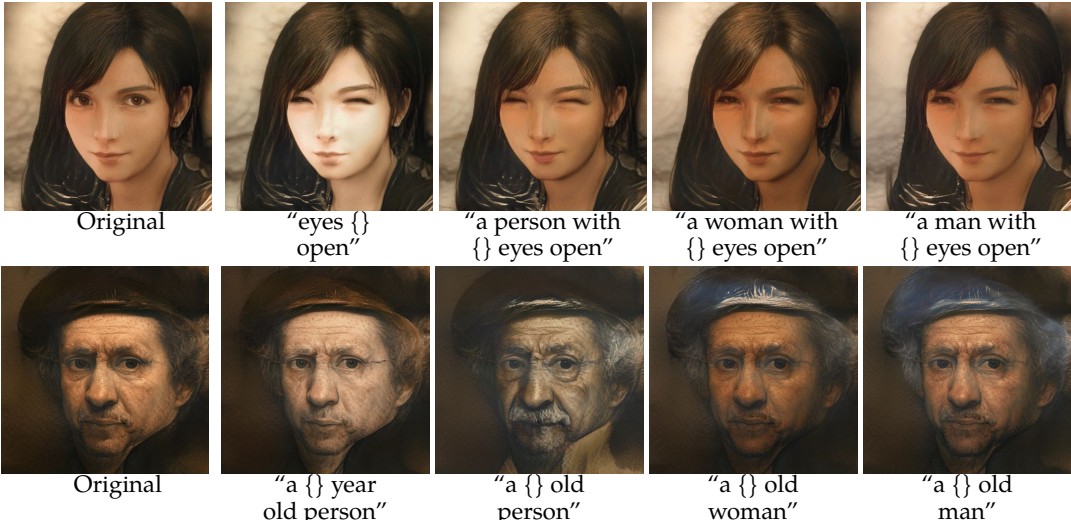

| Original | "eyes {} open" | "a person with {} eyes open" | "a woman with {} eyes open" | "a man with {} eyes open" |
|---|---|---|---|---|

| Original | "a {} year old person" | "a {} old person" | "a {} old woman" | "a {} old man" |
|---|---|---|---|---|

Figure 14: **Ablation on the choice of text template for micromotion "opening eyes" and "aging face".** For each template, we fill the wildcard "{}" using descriptive text, including {10%, 20%, ..., 100%}, {10, 20, ..., 60}, and {small, big, ...}. For conciseness, we only show the last frame of each group. Best view when zoomed in.

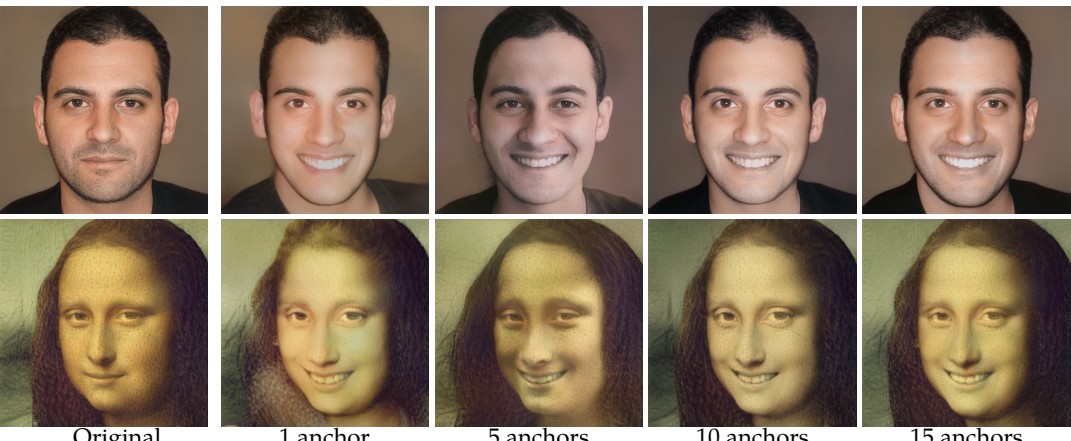

| Original | 1 anchor | 5 anchors | 10 anchors | 15 anchors |
|---|---|---|---|---|

Figure 15: **Ablation on the number of anchoring latent codes used to find the low-rank latent space.** We use the micromotion "smiling" as an example, while applying to both in-domain (human face) and out-of-domain (painting) images, we notice the quality of latent space editing improves proportionally w.r.t the number of anchors until it finally saturated at around 10 anchors.

space can sometimes be noisy, which introduces minor undesirable changes. Second, the training processes employed for StyleGAN-v3 and StyleGAN-XL may not be flawless, potentially leading to latent spaces with perturbations in the released checkpoints. We believe these issues can be alleviated with a larger set of reference latent codes and better-trained checkpoints.

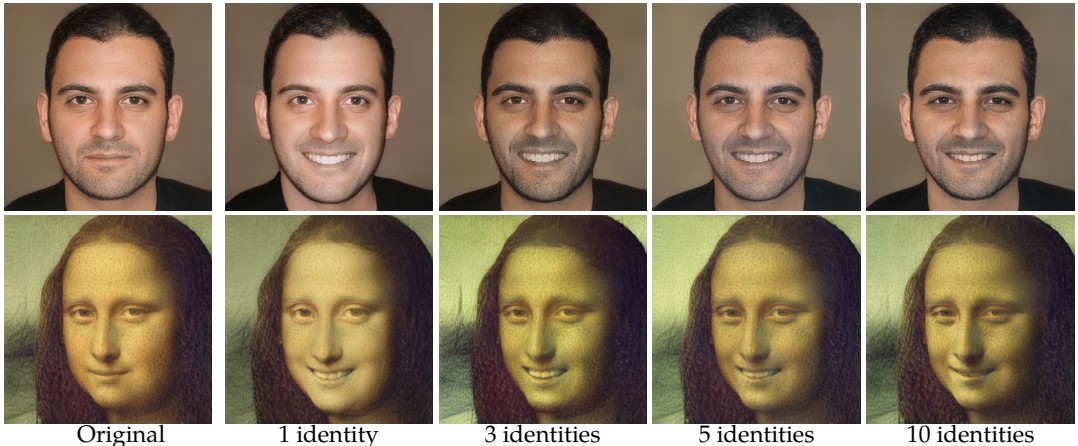

| Original | 1 identity | 3 identities | 5 identities | 10 identities |

Figure 16: **Ablation on the number of identities used in latent-space optimization.** We use the micromotion "smiling" as an example while applying to both in-domain (human face) and out-of-domain (painting) images. We notice there is no clear visual improvement in the quality of micromotion as the number of identities grows, however it would result in different styles of "smile".

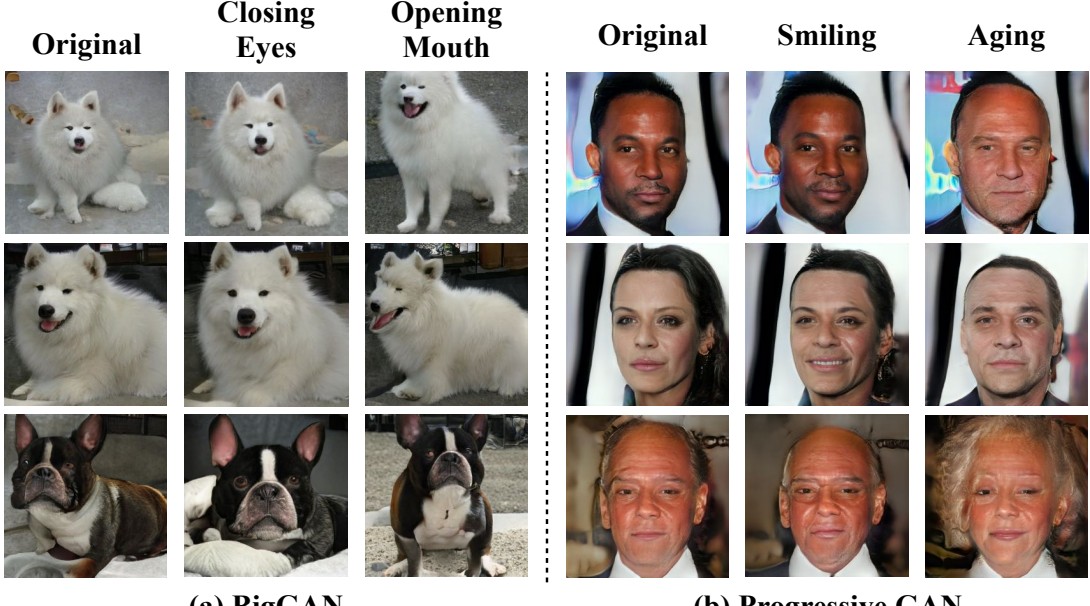

**(a) BigGAN**      **(b) Progressive GAN**

Figure 17: **Ablation on different GANs.** We demonstrate the editing results using different GANs, including BigGAN (left) and Progressive GAN (right). The target attributes are "closing eyes", "opening mouth" for dogs, and "smiling", "aging" for human faces.

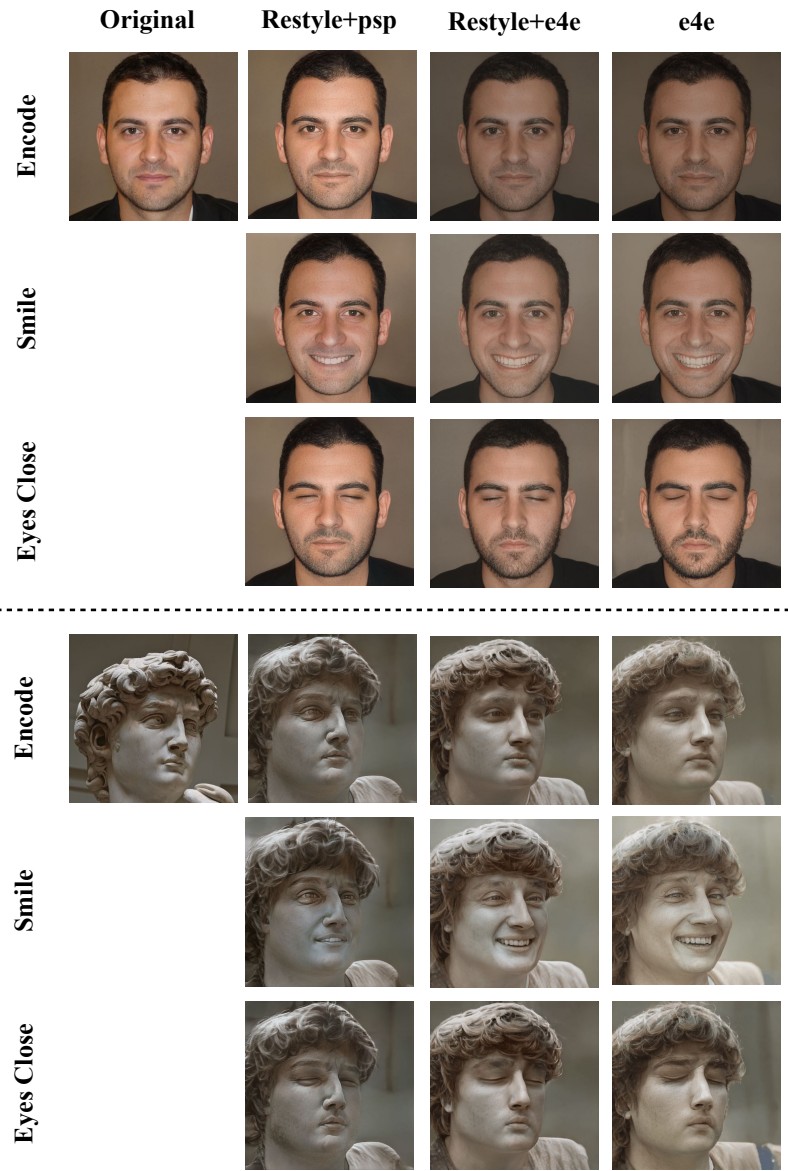

Figure 18: **Ablation on inversion methods.** We demonstrate editing results of using different inversion methods for both in-domain and out-of-domain input image.

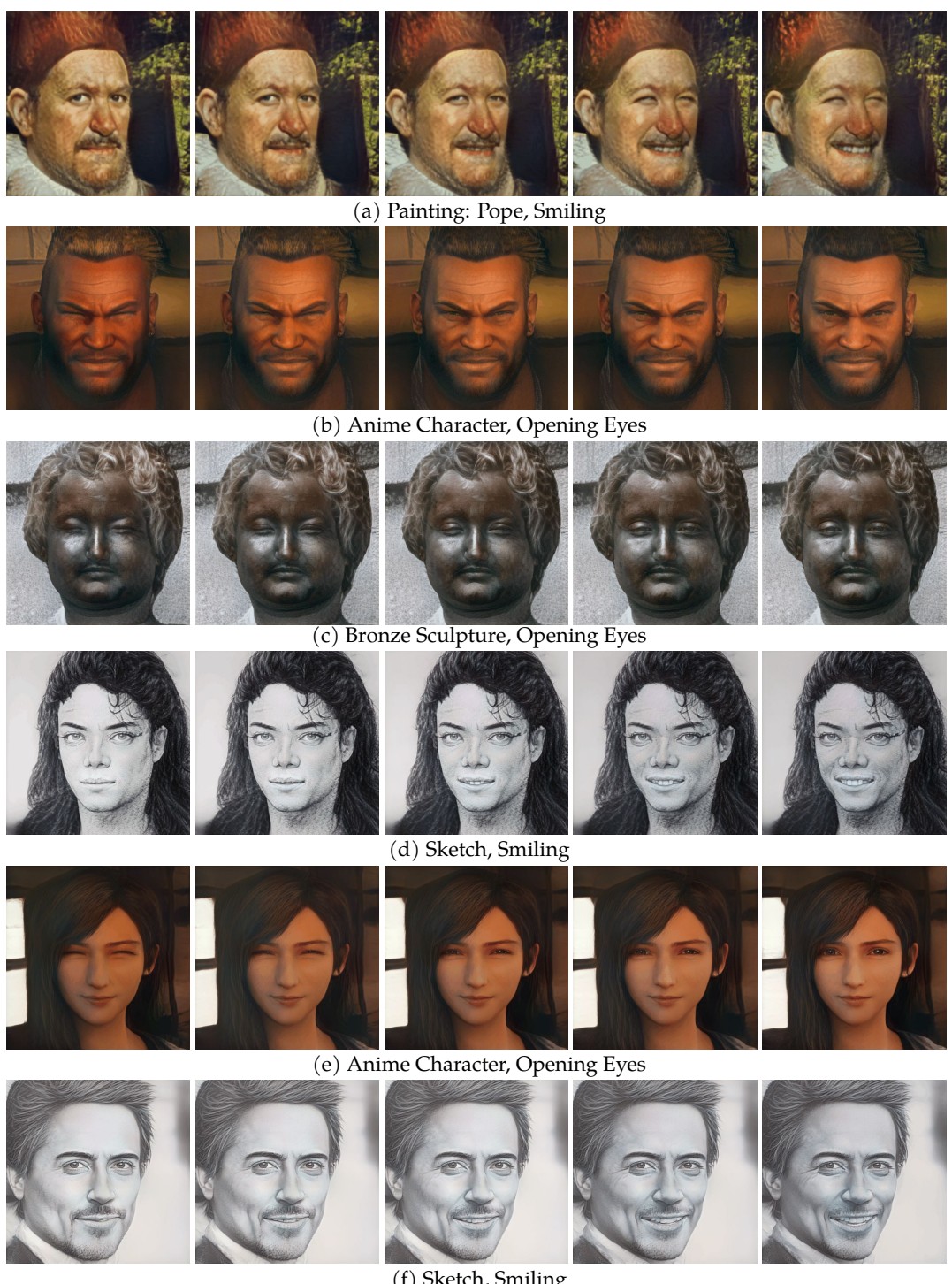

(a) Painting: Pope, Smiling

(b) Anime Character, Opening Eyes

(c) Bronze Sculpture, Opening Eyes

(d) Sketch, Smiling

(e) Anime Character, Opening Eyes

(f) Sketch, Smiling

Figure 19: **Additional examples of micromotions transferred to novel domains.** Our micromotions generalize well when transferred to novel domains, which include anime characters, sculptures, and various genres of paintings (oil painting, sketch). Best view when zoomed in.

| Original | Encode | Edit | Original | Encode | Edit |
|:---:|:---:|:---:|:---:|:---:|:---:|

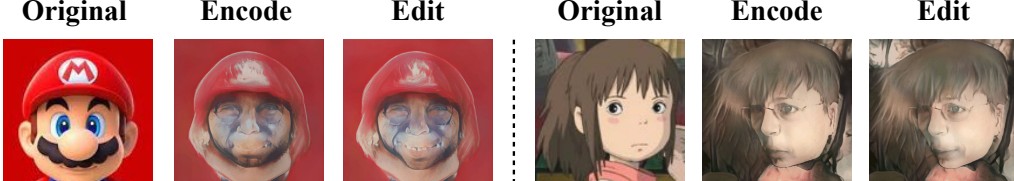

Figure 20: **Failure case study.** The target attribute is "smiling". We demonstrate that when the encoder fails to encode out-of-domain images (*e.g., Mario, Chihiro*), using the discovered editing direction will also synthesize incorrect image.

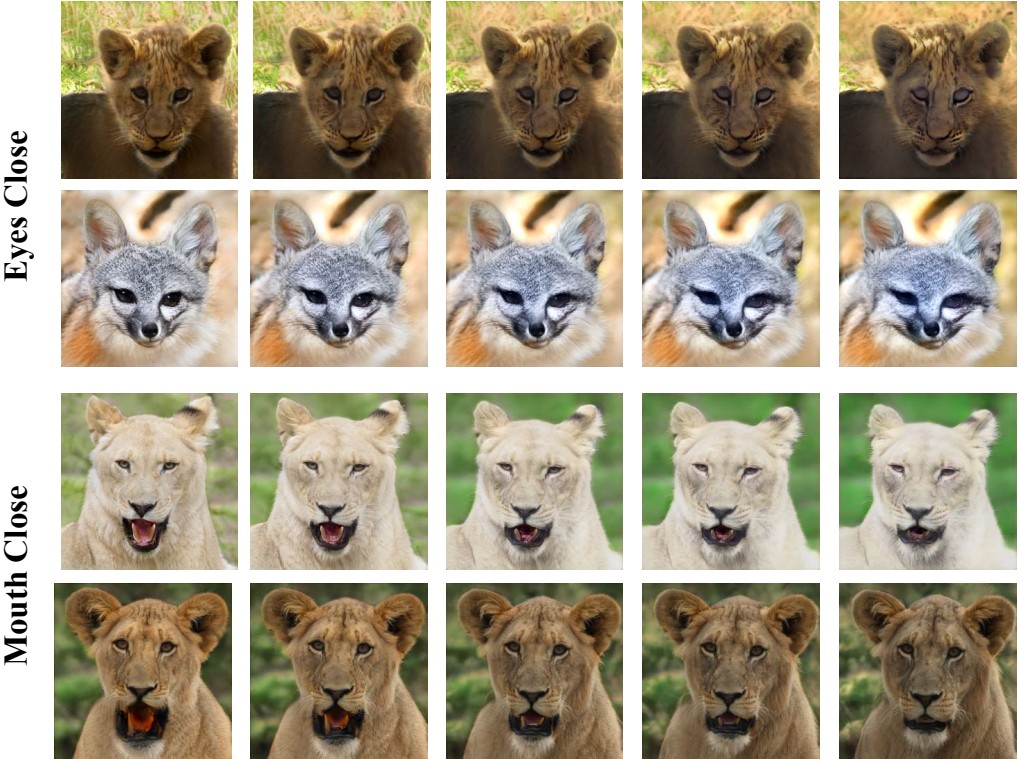

Figure 21: **Micromotions on wild animals.** The target micromotions are "eyes close" and "mouth close".

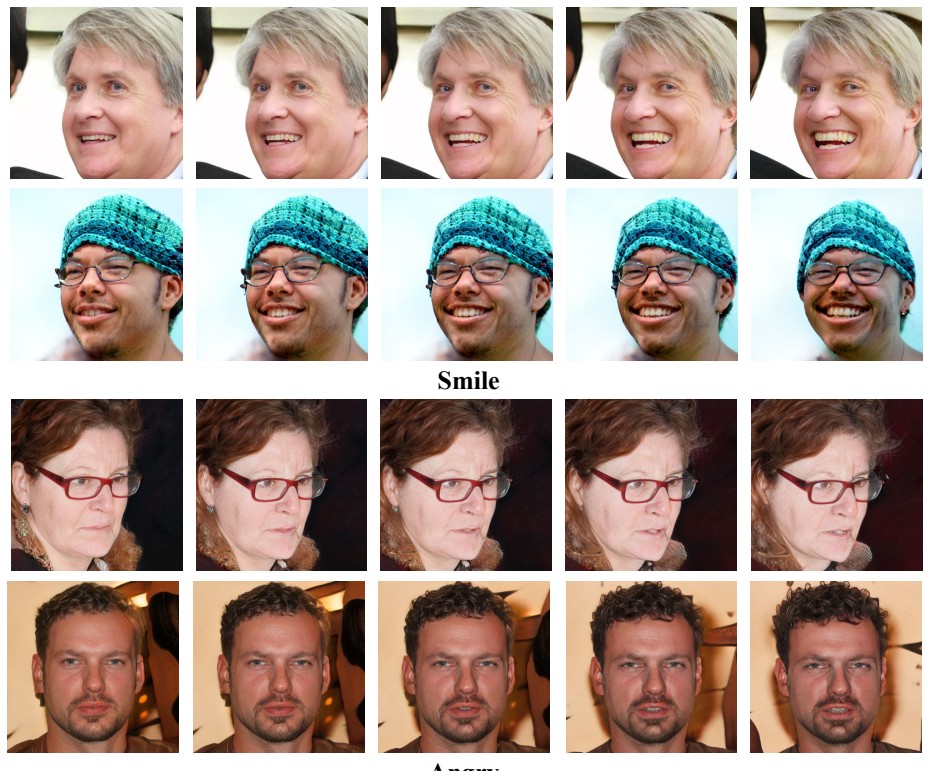

**Smile**

**Angry**

Figure 22: Micromotion examples found using StyleGAN-v3.

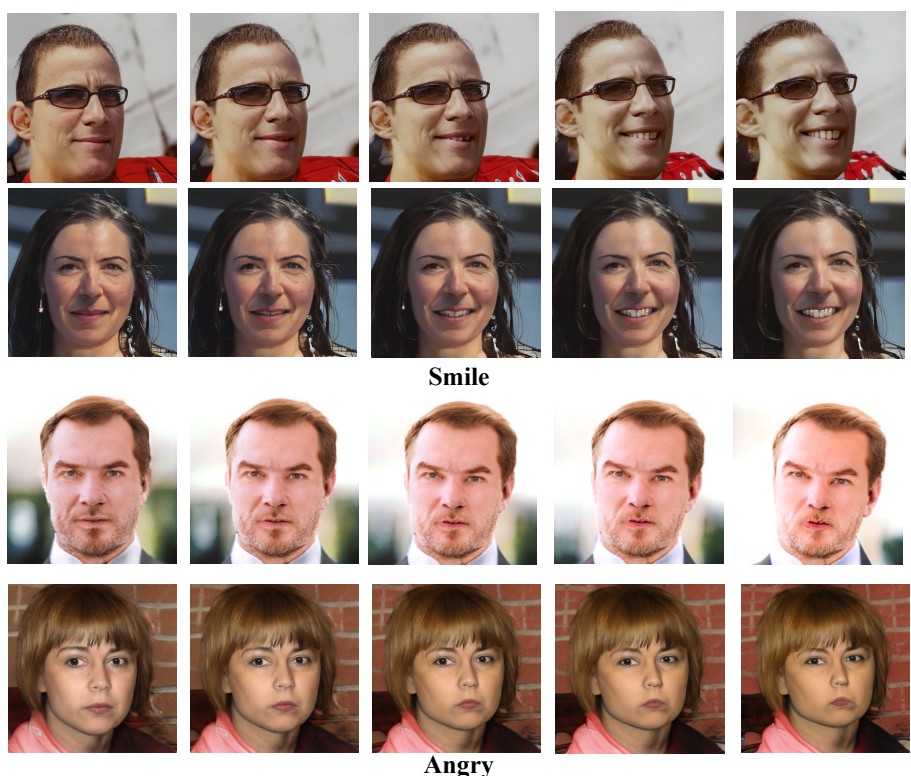

**Smile**

**Angry**

Figure 23: Micromotion examples found using StyleGAN-XL.

