# OpenReview forum: "Decoding Micromotion in Low-dimensional Latent Spaces from StyleGAN"
_CPAL.cc/2024/Conference — CPAL 2024 (Proceedings Track) Oral_

### Official Review · Reviewer_rDKo · 2023-10-06
**Good and interesting paper.**

**Rating:** 8
**Confidence:** 1

**Review:**

This paper proposed a method that utilizes the potential capacity of pre-trained StyleGAN to generate temporal micromotion frame sequences. Based on the hypothesis of Low-rank Micromotion Subspace, the workflow consists of three steps including Reference Anchoring which obtains a set of latent codes corresponding to the desired action performed by the same person, Robust space decomposition that utilizes robust PCA to find the edition direction, and subspace transformation that interpolate or extrapolate along the edit direction to obtain the intermediate frames.


The idea is clear and interesting. The results seem to be promising. The hypothesis is meaningful. I do recommend to accept this paper.

A question: Why the results of PCA could provide the edit direction \delta V? Could provide more detail about this? Moreover, what is the shape of \delta V?

---

### Official Review · Reviewer_i4G4 · 2023-10-07
**Valuable problem, good idea with interesting results**

**Rating:** 7
**Confidence:** 3

**Review:**

This paper studies the latent space of StyleGAN, a family of GAN-based generative models. Attributes editing by modifying the latent code of StyleGAN is an important task while existing works can hardly edit an attribute without other undesired changes. This paper tries to understand if this is because of the intrinsic limits of the entangled latent space, or, just because the existing works are not good enough at disentangling. They hypothesize that a low-rank feature space can be extracted from the StyleGAN high-dimensional feature space, where universal editing directions can be reconstructed from *micromotions*. Empirical results verify this hypothesis and show that the low-rank subspace can be used for high-quality editing.

Pros:

1. GAN-based generative models have shown impressive results in generation and editing. The understanding of it is relatively under-explored. This paper adds new insights in this track.

2.  The proposed low-rank subspace analysis is technically sound and interesting to me.

3. Empirically, the results support the proposed hypothesis and show better results of using it for high-fidelity editing over other approaches.

4. The paper is well-written, logic clear and neat.

Cons:

1. One concern is that only StyleGAN-v2 is evaluated, while the paper actually attempts to answer a question for the general StyleGAN family. I wonder if the conclusions found on StyleGAN-v2 can generalize to other StyleGAN models.

---

### Official Review · Reviewer_GYmM · 2023-10-16
**Decoding Micromotion in Low-dimensional Latent Spaces from StyleGAN**

**Rating:** 7
**Confidence:** 4

**Review:**

This paper explores the intrinsic properties of GAN hermitian spaces. The authors demonstrate that subtle movements can be represented in low-rank spaces derived from StyleGAN latent space. These micromotion features are decoded using short text or video clips as reference points. Besides, this micromotion knowledge can be transferred to different face images, even from diverse domains like paintings and sculptures. Yet, this work is great, a few questions arise:
(1). First of all, the authors have only experimented on StyleGAN-V2, and I think the generalizability of the method should be verified on more StyleGAN families, perhaps even on diffusion models.
(2). The article shows extraordinary generation results that demonstrate excellent micromotion decoupling and manipulation capabilities, but there is a little bit of blurring on the generated images in Figure.5 as well as in the upper part of the figure in Figure.7(a), Is this imperfection a natural outcome of the method, or could there be room for refinement?
(3). The experiments and supplementary experiments show some remarkably good generated results, but the presentation of the results in Figure.11 seems to be a bit blurry and distorted, could it be replaced with a clearer one?

---

### Meta-Review · Area_Chair_UDD4 · 2023-11-07

**Recommendation:** Accept (Poster)
**Confidence:** 3

**Metareview:**

The paper makes the claim that manipulations on low rank projections of the style-GAN latent space can produce disentangled effects. Even though the effect is not fully expected theoretically, the paper produces good empirical validation and promises to release the code.
I recommend to accept the paper, conditioned on the authors publishing of the code such that all the experiments are reproducible.

---

### Decision · Program_Chairs · 2023-11-19

**Decision:**

Accept (Oral)

**Comment:**

All reviewers and AC agreed that the paper is of high quality, this paper studies the latent space of StyleGAN and demonstrate that low rank projections of the style-GAN latent space can produce interesting disentangled effects. The work highly aligns with the theme of the conference, and presents very interesting results. The paper produces good empirical validation, and please release the code for reproducibility of the results.

The action PC chair for this paper is Qing Qu, who made the decision after carefully reading the paper as well as the comments by all reviewers and AC. The decision is agreed upon by all PC chairs.